# The Timing Effects of Soy Protein Intake on Mice Gut Microbiota

**DOI:** 10.3390/nu12010087

**Published:** 2019-12-27

**Authors:** Konomi Tamura, Hiroyuki Sasaki, Kazuto Shiga, Hiroki Miyakawa, Shigenobu Shibata

**Affiliations:** 1Laboratory of Physiology and Pharmacology, School of Advanced Science and Engineering, Waseda University, Shinjuku-ku, Tokyo 162-8480, Japan; tmr-knm@akane.waseda.jp (K.T.); hiroyuki-sasaki@asagi.waseda.jp (H.S.); k_shiga@fuji.waseda.jp (K.S.); hgbbst-hiroki@toki.waseda.jp (H.M.); 2National Institute of Advanced Industrial Science and Technology, AIST-Waseda University Computational Bio Big-Data Open Innovation Laboratory (CBBD-OIL), Shinjuku-ku, Tokyo 169-8555, Japan

**Keywords:** soy protein, microbiota, lipid metabolism, circadian, chrono-nutrition

## Abstract

Soy protein intake is known to cause microbiota changes. While there are some reports about the effect of soy protein intake on gut microbiota and lipid metabolism, effective timing of soy protein intake has not been investigated. In this study, we examined the effect of soy protein intake timing on microbiota. Mice were fed twice a day, in the morning and evening, to compare the effect of soy protein intake in the morning with that in the evening. Mice were divided into three groups: mice fed only casein protein, mice fed soy protein in the morning, and mice fed soy protein in the evening under high-fat diet conditions. They were kept under the experimental condition for two weeks and were sacrificed afterward. We measured cecal pH and collected cecal contents and feces. Short-chain fatty acids (SCFAs) from cecal contents were measured by gas chromatography. The microbiota was analyzed by sequencing 16S rRNA genes from feces. Soy protein intake whether in the morning or evening led to a greater microbiota diversity and a decrease in cecal pH resulting from SCFA production compared to casein intake. In addition, these effects were relatively stronger by morning soy protein intake. Therefore, soy protein intake in the morning may have relatively stronger effects on microbiota than that in the evening.

## 1. Introduction

Mammals have approximately 100 trillion bacteria in their gut that comprise the microbiota. Gut microbiota has profound influences on the host’s physiological conditions such as nutrient absorption, metabolism, and immunity [1]. Microbial alterations cause inflammatory bowel diseases and metabolic disorders. For example, concerning the microbiota of an obese person, the relative abundance of *Firmicutes*, which is a factor of obesity, is increased and *Bacteroidetes*, which prevents fat accumulation, is decreased [2]. In addition, an altered microbiota causes obesity because germ-free mice show an increase in body fat when injected with the microbiota of an obese mice [3].

Intestinal bacteria digest non-digestible food components such as dietary fibers, oligosaccharides, resistant starches, and resistant proteins, and produce short-chain fatty acids (SCFAs) [4,5]. SCFAs include acetic acid, propionic acid, lactic acid, and butyric acid. SCFAs are used as an energy source for colonic epithelial cells [6]. SCFAs maintain gut acidic conditions and prevent the growth of harmful bacteria such as *Enterobacteriaceae* and *Clostridia* [4,7]. SCFAs also have beneficial effects on mammalian energy metabolism and regulate the metabolism of fatty acid, glucose, and cholesterol [4].

The alteration of microbiota depends on various factors such as age, stress, disease, drugs, and diet [8]. There are many reports about the relationship between diet and microbiota. The microbiota can be rapidly affected by dietary changes [9]. Some studies evaluated the impact of protein on the microbiota. Many proteins are absorbed by the small intestine. However, segments of some proteins pass through the small intestine to reach the large intestine [10]. These resistant proteins and amino acids are metabolized by intestinal microbiota to SCFAs [11]. For example, soy protein intake causes higher microbial diversity and SCFA levels [12,13].

The circadian clock system plays an important role in maintaining physiological conditions such as the sleep-wake cycle, body temperature, and metabolism [14]. In the mammalian circadian system, there are two clocks: the main central oscillator in the suprachiasmatic nuclei (SCN), and the peripheral oscillator in peripheral organs. The SCN clock is mainly entrained by light-dark stimuli, and it regulates the peripheral clocks. The peripheral clocks are entrained by pharmacological agents, food nutrients, and mental or physical stress [15,16,17,18,19]. The SCFAs produced by the microbiota also entrain the circadian clock [20]. Microbiota exhibits diurnal oscillations in composition and function in both mice and humans. In addition, jet lag induced circadian disruption changes microbiota, and when feces from jet lag mice were transferred to germ-free mice, they became obese [21]. Thus, the activity of microbiota is strongly associated with circadian rhythm.

Chrono-nutrition is the science of nutrition, which is based on chronobiology. Hormonal secretion and the metabolism and absorption of nutrients have circadian variations [16]. Therefore, some food components have the most effective intake timing [22,23]. For example, fish oil intake in the morning rather than the evening is more effective to reduce lipids in mice [23]. Intake of water-soluble dietary fiber in the morning has a greater effect on microbiota diversity rather than in the evening [24]. In this study, we examined the effective timing of soy protein intake. Soy protein, especially β-conglycinin, is known to have beneficial effects on hepatic lipid metabolism, prevention of hepatic steatosis, and reduction of body fat in both rodents and humans [25,26,27,28,29]. It is also reported that soy protein intake has a superior effect on microbiota [12,26,27]. As hormonal secretion as well as microbial composition and function exhibit diurnal oscillations, the effective timing to alter microbiota can be different according to the food components [14,21]. However, the effective timing of soy protein intake has not been investigated. Therefore, in this study, we examined the effect of soy protein intake on mice gut microbiota based on chrono-nutrition, such as morning intake or evening intake.

## 2. Materials and Methods

### 2.1. Animals and Diets

We used 105 of ICR 8-week-old male mice (Tokyo Laboratory Animals, Tokyo, Japan) in this study. The mice were kept under 12 h light/12 h dark condition. Lights-on time was defined as zeitgeber time 0 (ZT0) and lights-off time as ZT12. Each mouse was housed in a plastic cage individually, at a temperature of 22 ± 2 °C, humidity of 60% ± 5%, and light intensity of 100-150 lux. We prepared two kinds of diet, a high-fat diet (HFD) with casein and HFD with soy protein (Fujipro F, Fuji Oil Co., Osaka, Japan) (Table 1).

To produce metabolic syndrome models with obesity, high inflammation, and abnormal microbiota, the mice were fed a HFD with casein and water ad libitum for one week before commencing the experiments. Thereafter, mice were fed HFD with casein or soy protein according to the experimental protocols. The Committee for Animal Experimentation at Waseda University approved all experimental protocols (permission protocol 2018-A030).

### 2.2. Experimental Design

In this study, we prepared two kinds of diets, HFD with casein (casein diet) and HFD with soy protein (soy diet), as described previously. For microbiota deterioration, mice were fed a casein diet for one week before commencing the experiments.

In experiment 1, we examined the effects of soy protein intake in a short period. Soy protein is known to have anti-obesity effects. Obesity causes a change in microbiota composition. Therefore, to eliminate the effect of body weight differences, we conducted the experiments over a short period, before body weight change. Mice were given free access to the casein diet (Casein group) or soy diet (Soy group) for 10 days, and they were sacrificed at ZT12, ZT20, or ZT4. Ten mice were prepared for each time point and group. We measured the cecal pH, and we collected the cecal contents, feces, blood, and liver samples (Figure 1a).

In experiment 2, we examined the timing effects of soy protein intake. To compare the effect of soy protein intake in the morning to that in the evening, mice were fed 1.8 g diets twice a day in the morning (ZT12) and evening (ZT20). When mice were given 4 h of access to food in the morning and evening, the amount of food consumption was different between the morning and evening [24]. Therefore, we fed mice 1.8 g diets twice a day, so that morning and evening food consumption is the same. It is reported that mice were able to consume all of 1.8 g diet within 4 h [30,31]. The experiment period was set to 14 days since it takes a few days for mice to adapt to the 2-meals-per-day feeding pattern. The mice were fed only casein diet (Casein group), soy diet in the morning and casein diet in the evening (M-Soy group), or casein diet in the morning and soy diet in the evening (E-Soy group). The mice were kept under experimental conditions for two weeks, and then they were sacrificed at ZT12, 20, or 4. Five mice were prepared for each time point and group. We measured the cecal pH and collected the cecal contents, feces, blood, and liver samples (Figure 1b).

### 2.3. Cholesterol and Triglyceride Measurement

Serum cholesterol and triglyceride (TG) levels were measured using cholesterol and triglyceride kit (FUJIFILM Wako Pure Chemical Co., Osaka, Japan). The assay was performed according to the manufacturer’s instructions.

### 2.4. Real-Time RT-PCR

Relative liver mRNA levels were measured by real-time RT-PCR. The mice were anesthetized with isoflurane and sacrificed at ZT 12, ZT 20, or ZT4. We collected livers to measure mRNA levels at each time point. Total liver RNA was extracted using RNA-*Solv* Reagent (Omega Bio-Tek Inc., Norcross, GA, USA). RNA concentration of each sample was adjusted using a spectrophotometer (GE Healthcare Japan Co., Tokyo, Japan). The RNA was reverse-transcribed and amplified using One-Step SYBR RT-PCR kit (Takara Bio Inc., Shiga, Japan) with specific primer pairs (Table 2) on Piko Real PCR system (Thermo Fisher Scientific, Waltham, MA, USA). The relative expression levels of target genes were normalized with *GAPDH*. The data were analyzed using the ΔΔCt method.

### 2.5. Cecal pH Measurement

Cecal pH was measured using pH meter (Euthech Instruments, Vernon Hills, IL, USA). The electrode of the pH meter was inserted directly into the cecum, immediately after collection.

### 2.6. Short-Chain Fatty Acid (SCFA) Measurement

Short-chain fatty acid (SCFA) in cecal contents was measured through gas chromatography (Shimadzu Co., Kyoto, Japan) as described in a previous report [32]. Cecal contents were acidified with sulfuric acid and SCFAs were extracted from 50 mg of cecal contents by shaking in 50 μL of sulfuric acid, 400 μL of diethyl ether, and 200 μL of ethanol (FUJIFILM Wako Pure Chemical Co., Osaka, Japan). The mixture was centrifuged at 18700× *g* for 30 s. The supernatant (1 μL) was injected into the capillary column (InertCap Pure WAX (30 m × 0.25 mm, df = 0.5 μm), GL Science, Tokyo, Japan) of gas chromatography coupled to a flame ionization detector. The initial temperature was 80 °C and the final temperature was 200 °C. Helium was used as carrier gas and quantification of the samples was performed using calibration curves for acetic, lactic, propionic, and butyric acids. A standard curve plotted for the quantitation of each acid was in the samples.

### 2.7. Fecal DNA Extraction

Fecal DNA was extracted according to the previous report with modifications [33]. We collected feces from the rectum, when we sacrificed the mice at each time point. Approximately 0.2 g fecal sample was suspended in a 50 mL tube containing 20 mL PBS. The suspension was filtered through a 100 μm nylon filter (Corning Inc., New York, NY, USA). The tube was washed with 10 mL PBS and then filtered through the filter. The filtrates were centrifuged at 9000× *g* for 20 min at 4 °C, and the supernatants were removed. Each precipitate was suspended in 1.5 mL TE 10 buffer (10 mM Tris-HCl (FUJIFILM Wako Pure Chemical Co., Osaka, Japan)/10 mM EDTA (DOJINDO, Tokyo, Japan)), and the suspension was transferred to 2 mL microtube. The suspensions were centrifuged at 9560× *g* for 5 min at 4 °C, and the supernatants were removed. Each precipitate was suspended in 800 μL TE 10 buffer. The suspensions were added 100 μL lysozyme (150 mg/mL) (FUJIFILM Wako Pure Chemical Co., Osaka, Japan) and then incubated for 1 h at 37 °C. Achromopeptidase (20 μL, 100 units/μL, FUJIFILM Wako Pure Chemical Co., Osaka, Japan) was added to the suspension and then incubated for 30 min at 37 °C. The suspension was treated with 50 μL of 20% sodium dodecyl sulfate and proteinase K (Promega Co., Madison, WI, USA) and then incubated for 1 h at 55 °C. To extract DNA, 980 μL PCI (phenol/chloroform/isoamyl alcohol) (Invitrogen, Carlsbad, CA, USA) was added and centrifuged at 6000× *g* for 10 min at 20 °C. The supernatant was transferred to a new 2 mL microtube and then suspended with 100 μL of 3 M sodium acetate and 900 μL isopropanol (FUJIFILM Wako Pure Chemical Co., Osaka, Japan). The suspensions were centrifuged at 6000× *g* for 10 min at 20 °C, and the supernatants were removed. The DNA pellet was rinsed with 1 mL of 70% ethanol and dried. The DNA was purified by treatment with 99 μL TE buffer and 1 μL RNase (10 μg/mL) (FUJIFILM Wako Pure Chemical Co., Osaka, Japan), then precipitated with 100 μL of 20% PEG solution (TOKYO Chemical Industry Co., Tokyo, Japan). The DNA was pelleted by centrifugation at 10,000× *g* at 4 °C, rinsed with 500 μL of 70% ethanol, and dissolved in 50 μL TE buffer.

### 2.8. 16S rRNA Gene Sequencing

16S rRNA gene sequencing was performed according to Illumina instructions (16S Metagenomic Sequencing Library Preparation). V3-V4 variable regions of the 16S rRNA gene were amplified using forward primer (5′-TCGTCGGCAGCGTCAGATGTGTATAAGAGACAGCCTACGGGNGGCWGCAG-3′) and reverse primer (5′-GTCTCGTGGGCTCGGAGATGTGTATAAGAGACAGGACTACHVGGGTATCTAATCC-3′). Amplicon PCR was performed with 2.5 μL microbial DNA (5 ng/μL), 5 μL of each amplicon PCR primer (1 μM), and 12.5 μL of 2 × KAPA HiFi HotStart Ready Mix (Kapa Biosystems, Wilmington, MA, USA) under conditions of 3 min at 95 °C, 25 cycles of 95 °C for 30 s, 55 °C for 30 s, and 72 °C for 30 s, and a final extension of 72 °C for 5 min. The PCR products were purified using AMPure XP beads (Beckman Coulter, Inc., Brea, CA, USA).

The Nextera XT Index Kit v2 (Illumina Inc., San Diego, CA, USA) was used to join dual indices and Illumina sequencing adapters. Index PCR was performed in 5 μL PCR production, 5 μL of each Nextera XT Index primer, 25 μL of 2 × KAPA HiFi HotStart Ready Mix, and 10 μL of PCR Grade water under conditions of 3 min at 95 °C, 8 cycles of 95 °C for 30 s, 55 °C for 30 s, and 72 °C for 30 s, and a final extension of 72 °C for 5 min. The PCR products were purified using AMPure XP beads. The quality of the purification was checked using Agilent 2100 Bioanalyzer with DNA 1000 kit (Agilent Technologies, Santa Clara, CA, USA). Finally, the DNA library concentration was diluted to 4 nM.

The DNA library was sequenced using Miseq reagent kit v3 (Illumina Inc., San Diego, CA, USA) on Illumina Miseq 2 × 300 bp platform. This sequencing was performed following manufacturer instructions.

### 2.9. Analysis of 16S rRNA Gene Sequences

16S rRNA gene sequence reads were processed through quantitative insights into microbial ecology (QIIME) pipeline version 1.9.1 [34]. Quality-filtered sequence reads were assigned to operational taxonomic units (OTUs) at 97% identity with the UCLUST algorithm [35]. These reads were then compared to the reference sequence collections in the Greengenes database (August 2013 version). A total of 4,034,110 reads were obtained from 105 samples. On average, 38,420 ± 2427 reads were obtained per sample. Taxonomy summary, alpha-diversity (within-sample), beta-diversity (between-sample dissimilarity), and principal coordinate analysis (PCoA) were calculated and generated by QIIME. PCoA analysis was also calculated using unweighted UniFrac distances.

### 2.10. Metagenome Prediction

The functional profiles of microbial communities were predicted through phylogenetic investigation of communities by reconstruction of unobserved states (PICRUSt) [36]. The functional predictions were assigned to almost all Kyoto Encyclopedia of Genes and Genomes (KEGG) ortholog (KO) functional profiles of microbial communities via 16S sequences. We selected and examined categories related to “Amino Acid Metabolism” and “Energy Metabolism” for analysis of simplification and clarity.

### 2.11. Statistical Analysis

Data are expressed as mean ± standard error of the mean (SEM). In this study, we compared the feeding condition at each time point, because we focused on the difference in feeding condition rather than time point. Statistical analysis was performed using GraphPad Prism version 6.03 (GraphPad Software, San Diego, CA, USA). The data were tested for normality and equality of variances using a D’Agostino-Pearson test/Kolmogorov-Smirnov test and Bartlett’s test, respectively. Parametric analysis was conducted using one-way ANOVA with Tukey test or Student’s *t*-test for post-hoc analysis, and non-parametric analysis was conducted using the Kruskal-Wallis test with Dunn’s test or the Mann-Whitney test for post-hoc analysis. The differences in microbiota composition were tested using the permutational multivariate analysis of variance (PERMANOVA). PERMANOVA was analyzed by QIIME.

## 3. Results

### 3.1. Soy Protein Intake Affected Lipid Metabolism and the Gut Microbiota

It has already been reported that soy protein not only reduces serum cholesterol and triglycerides, but also changes the microbiota composition, leading to considerable microbial diversity [12,25,26]. To examine whether the results of our study are similar to previous reports, we considered the effect of soy protein feeding in the free-feeding condition.

First, we examined the effect of soy protein on lipid metabolism. The food consumption (Casein group: 3.88 ± 0.15 g/day, Soy group: 3.92 ± 0.11 g/day) and final body weight (Casein group: 42.69 ± 0.59 g, Soy group: 42.15 ± 0.71 g) showed no differences between the groups. We showed the data of each time point and the average of a 3-time point (AVE). The serum cholesterol of the Soy group was significantly lower than that of the Casein group at ZT20 and AVE. Serum TG level showed no significant difference between both groups (Figure 2a). We measured the mRNA expression levels of fatty acid and cholesterol metabolism-related genes from liver samples. *Acc1*(ZT12, ZT20, and AVE), *Fasn* (ZT12, ZT20, and AVE), and *Srebp1c* (ZT12, ZT4, and AVE) expression levels were significantly lower in the Soy group than those in the Casein group. *Cyp7α1* expression level tended to be higher in the Soy group than that in the Casein group at ZT20 (Figure 2b).

To examine the effect of soy protein on microbiota, we measured cecal pH and SCFA production. Cecal pH was significantly lower in the Soy group than that in the Casein group (Figure 3a). Acetic acid (ZT12, ZT4, and AVE), propionic acid (ZT12 and ZT4), lactic acid, and butyric acid levels were significantly or tended to be higher in the Soy group than those in the Casein group (Figure 3b).

As cecal pH was decreased and SCFA production was increased, soy protein intake may alter the microbiota. Therefore, we analyzed the microbiota from feces. The Soy group showed significantly higher alpha-diversity for the Simpson index than that in the Casein group at ZT20 (Figure 4a). The PCoA of unweighted UniFrac distance showed that the beta-diversity of microbiota composition was significantly different between the Soy group and the Casein group (Figure 4b). Concerning the relative abundances of microbes at the phylum level, *Bacteroidetes* (ZT20) and *Proteobacteria* (ZT12, ZT20, and AVE) in the Soy group were significantly higher than those in the Casein group. *Firmicutes* (ZT12, ZT20, and AVE) in the Soy group were significantly lower than those in the Casein group (Figure 4c). At the genus level, *Bifidobacterium* (ZT12, ZT4, and AVE), *Enterococcus* (ZT20 and AVE), *[Ruminococcus]* (ZT20 and AVE), and *Desulfovibrio* (ZT20 and AVE) in the Soy group were significantly higher than those in the Casein group. *Lactococcus* in the Soy group was significantly lower than that in the Casein group (Figure 4d).

To infer the metagenome functional content based on the microbial community profiles obtained from 16S rRNA gene sequences, we used PICRUSt. The microbial communities could be distinguished based on their functions. The KEGG pathways associated with amino acid and energy metabolisms were significantly upregulated in the Soy group. The pathways associated with glycine, serine, and threonine metabolisms (ZT12, ZT20, and AVE) and lysine biosynthesis (ZT20 and AVE) in the Soy group were significantly upregulated compared to those in the Casein group (Figure 5a). The pathways associated with methane metabolism (ZT12, ZT20, and AVE) and nitrogen metabolism (ZT12, ZT20, and AVE) in the Soy group were significantly upregulated compared to those in the Casein group (Figure 5b).

### 3.2. Soy Protein Intake in the Morning Affected the Gut Microbiota More Than That in the Evening

Since soy protein intake improved microbiota, we examined the effect of soy protein intake timing on microbiota. To compare the effect of soy protein intake in the morning and evening, mice were fed 1.8 g diets twice a day in the morning (ZT12) and evening (ZT20). Mice were fed only casein diet (Casein group), soy diet in the morning and casein diet in the evening (M-Soy group), or casein diet in the morning and soy diet in the evening (E-Soy group). Mice were kept under the experimental condition for two weeks and were then sacrificed at ZT12, 20 or 4 (Figure 1b). We measured cecal pH and collected cecal contents, feces, blood, and liver sample.

First, we examined the effect of soy protein intake timing on lipid metabolism. The final body weight (Casein group: 43.57 ± 0.76 g, M-Soy group: 43.86 ± 0.51 g, E-Soy group: 42.50 ± 0.77 g) showed no significant difference among the groups. We showed the data of each time point and the average of a 3-time point (AVE). Serum cholesterol in the M-Soy group was significantly higher than that in the Casein and the E-Soy groups at ZT12. The serum TG level showed no significant difference (Figure 6a). We measured the mRNA expression levels of fatty acid and cholesterol metabolism-related genes in the liver sample. *Acc1* expression level was significantly lower in the E-Soy group than that in the Casein group at ZT12 and AVE. *Fasn* expression level in the E-Soy group at ZT12 was significantly lower than that in the M-Soy group and tended to be lower than that in the Casein group. *Cyp7α1* expression level in the Casein group was significantly higher than that in the M-Soy group at ZT20 and ZT4, and that in the E-Soy group at ZT4 (Figure 6b).

To examine the effect of soy protein intake timing on microbiota, we measured cecal pH and SCFA production. Cecal pH in the M-Soy group tended to be lower than that in the Casein group at ZT12, and significantly lower than that in the other groups at ZT20. The E-Soy group showed a significantly lower pH than those in the other groups at ZT4. The M-Soy and the E-Soy groups showed significantly lower cecal pH than that in the Casein group on AVE (Figure 7a). The lactic acid in the M-Soy group was significantly higher than that in the Casein group and tended to be higher than that in the E-Soy group at ZT20. The lactic acid in the E-Soy group was significantly higher than that in the Casein group at ZT4. Only the M-Soy group showed a significantly higher level of lactic acid than that in the Casein group on AVE. Butyric acid in the M-Soy group tended to be higher than that in the Casein group at ZT20 and ZT4, and significantly higher than that in the E-Soy group at ZT20. Butyric acid in the E-Soy group was significantly higher than that in the other groups at ZT4. Only the M-Soy group showed a significantly higher level of butyric acid than that in the Casein group on AVE (Figure 7b).

As the cecal pH was decreased and SCFA production was increased, soy protein intake in the morning may strongly alter the microbiota. Therefore, we analyzed the microbiota from feces. The M-Soy group showed tend to higher alpha-diversity for Simpson index than that shown by the other groups on AVE (Figure 8a). The PCoA of unweighted UniFrac distance showed that the beta-diversity of microbiota composition was significantly different between the Casein and the M-Soy group at ZT20. On the other hand, between the Casein and the E-Soy group, the beta-diversity of microbiota composition didn’t show a significant difference at ZT4 (Figure 8b). The PCoA of unweighted UniFrac distance of all-time points was significantly different between the Casein and the M-Soy groups (statistic value = 2.478, *p* = 0.002), and relatively different between the Casein and the E-Soy groups (statistic value = 1.460, *p* = 0.048) (Figure 8c). For the relative abundance of microbes at the phylum level, *Bacteroidetes* was significantly higher in the M-Soy group than that in the Casein group on AVE. The relative abundance of *Firmicutes* was significantly lower in the E-Soy group than that in the Casein group at ZT20 and AVE (Figure 8d). In the genus level, the relative abundance of *Lactococcus* in the M-Soy group was significantly lower than that in the other groups at ZT20, and the relative abundance of *Lactococcus* in E-Soy group was significantly lower than that in the other groups at ZT4. On AVE, the relative abundance of *Lactococcus* in the M-Soy group was significantly lower and that in the E-Soy group tended to be lower than that in the Casein group. The relative abundance of *[Ruminococcus]* in the E-Soy group was significantly lower than that in the other groups at ZT20 (Figure 8e).

To infer the metagenome functional content based on microbial community profiles obtained from 16S rRNA gene sequences, we used PICRUSt. The microbial communities could be distinguished based on their functions. The KEGG pathways associated with glycine, serine, and threonine metabolism in the M-Soy group were significantly or tended to be upregulated at ZT12 and AVE compared to those in the Casein group (Figure 9a). The pathways associated with methane metabolism in the M-Soy group were significantly or tended to be upregulated compared to those in the Casein group at ZT12 and AVE, and compared to those in the E-Soy group at ZT20. The pathways associated with methane metabolism in the E-Soy group were significantly upregulated compared to those in the Casein group at ZT12. The pathways associated with nitrogen metabolism in the M-Soy group tended to be upregulated compared to those in the other groups on AVE (Figure 9b).

## 4. Discussion

In this study, 10 days of soy protein intake reduced serum cholesterol and fatty acid synthesis related genes expression levels were experimented with in mice. In addition, soy protein changed microbial conditions and decreased cecal pH caused by SCFA production. Two weeks of soy protein feeding in the morning or evening resulted in a decrease in cecal pH and an increase in SCFA and microbiota diversity change after soy protein intake. In addition, soy protein intake in the morning may have a longer effect on SCFA production and cecal pH reduction than that of soy protein intake in the evening. It was suggested that soy protein might attenuate abnormality in gut microbiota effectively when taken in the morning rather than in the evening.

In experiment 1, soy protein reduced serum cholesterol level and fatty acid synthesis related genes such as *Acc1*, *Fasn*, and *Srebp1c* expression levels (Figure 2). It has already been reported that soy protein reduces serum cholesterol, TG, and fatty acid synthesis related genes expression levels [12,26]. In these reports, the effects of long-term soy protein intake were examined. In our study, mice were fed soy protein only for 10 days. However, serum cholesterol and fatty acid synthesis related genes expression levels were decreased. It was reported that SCFAs produced by the microbiota upregulated the expression of GLP-1 via activation of the MAPK pathway. GLP-1 induced reduction in mRNA expression of the fatty acid synthesis related genes [37,38,39]. Therefore, it is suggested that SCFA production by soy protein intake may be related to a reduction in fatty acid synthesis related gene expression. Since the current study focused on the effects of soy protein on microbiota, we measured only mRNA levels but not protein levels of the fatty acid synthesis related genes. To support these mRNA data, it might be necessary to measure the protein levels of enzyme activity.

Soy protein also changed the microbiota composition (Figure 4). Soy protein intake enhanced the production of SCFA, especially lactic and butyric acids, and decreased cecal pH (Figure 3). Previous studies also reported that soy protein causes greater diversity of microbiota than milk protein does, and their microbiota showed different compositions [12,25,26,27]. However, these previous studies examined the effects of long-term soy protein intake with bodyweight changes. Therefore, these reports could not exclude the possibility that soy protein provides anti-obesity effects and then improves the microbiota. In the current study, only 10 days of soy protein intake changed the microbiota without body weight changes. Our previous study reported that water-soluble dietary fiber changed microbiota in 10 days [24]. It is suggested that soy protein intake itself changes the microbiota and even short-term intake is effective in changing the microbiota.

In the present study, the relative abundances of various bacteria were changed by soy protein intake both at the phylum and genus levels (Figure 4c,d). These results were similar to previous reports [27,40]. *Firmicutes* are known as the obese factor and its relative abundance is higher in obese people [2]. In this experiment, *Firmicutes* was decreased by soy protein intake. This result suggests that soy protein may have an anti-obesity effect by decreasing *Firmicutes*. *Bifidobacterium* and *Enterococcus* are known to produce acetic and lactic acids through fermentative metabolism [41,42,43]. An increase in the relative abundance of *Bifidobacterium* and *Enterococcus* might cause an increase in acetic and lactic acids. It has also been reported that oral administration of *Bifidobacterium breve* to infants may prevent digestive disease [44]. Thus, Soy protein intake may prevent diseases of the intestines by increasing the relative abundance of *Bifidobacterium*. These changes in microbiota may be related to a decrease in serum lipid and hepatic fatty acid synthesis related gene expression. *Lactococcus* is known to produce lactic acid. However, *Lactococcus* was decreased by soy protein intake in this study. The reason may be because *Lactococcus* is the bacteria commonly found in raw milk, cheese, and other dairy products [45]. In this experiment, *Proteobacteria* and *Desulfovibrio* were increased by soy protein intake. It was reported that an increase in the abundance of *Proteobacteria* can change microbiota [46]. Further, *Desulfovibrio* in *Proteobacteria* can induce barrier dysfunction [47]. In the present study, we observed a greater diversity in the Soy group. However, some negative bacteria were increased. The reason underlying this increase in bacterial number and the associated mechanism under soy protein feeding condition remain unclear. In the future, this may be clarified by examining in detail the relationship between diet and microbiota.

In experiment 2, we examined the effect of soy protein intake timing (morning or evening) on microbiota. There were smaller effects of the diets on serum lipid, hepatic fatty acid, and cholesterol metabolism-related gene expression levels in experiment 2 than those in experiment 1 ( Figure 2; Figure 6), because the amount of soy protein intake was smaller in experiment 2 than in experiment 1. In addition, it has been reported that mice under time-restricted feeding of 8 h per day were protected against obesity and hepatic steatosis, with improved energy expenditure [48]. In this experiment, mice were restricted in not only the amount but also the timing of feeding. Therefore, the feeding schedule itself may have powerfully reduced the serum lipid levels and hepatic gene expression of the fatty acid synthesis related genes, and then this protocol may mask the effects of soy protein.

At first, we compared the effects of soy protein on the microbiota at 8 h after feeding initiation in both the groups (ZT20 for the morning intake and ZT4 for the evening intake), because microbiota was altered shortly after water-soluble dietary fiber intake under two-meals-per-day schedule [24] and rapidly affected by dietary changes [9]. The cecal pH and the amount of lactic acid and butyric acid showed similar effects in both morning and evening groups (Figure 7). These data suggest that soy protein has beneficial effects on the microbiota in the morning intake or evening intake as compared to the casein intake. In addition, the previous feeding has a strong effect on microbial conditions. Since we collected samples every 8 h in this study, we can discuss the effects throughout the day by taking the average of 3-time points. The morning soy protein intake also showed lower cecal pH before soy protein intake (ZT12) and higher levels of lactic and butyric acids at a one day average (Figure 7). Overall, it is suggested that soy protein intake in the morning may have long term effects on SCFA production and cecal pH reduction than that of the soy protein intake in the evening. Regarding microbiota, soy protein intake in the morning, not in the evening caused greater diversity on one day average (Figure 8a). In addition, microbiota varied to a greater extent by soy protein intake in the morning than that in the evening, shortly after soy protein intake (Figure 8b), and throughout the day (Figure 8c).

We do not know the detailed mechanism of such different effects on microbiota between morning intake and evening intake groups. The difference between soy protein intake in the morning and evening may be caused by a difference in fasting time before each diet. Two-meal-per-day feeding conditions were set close to a human feeding pattern in this study. In general, the fasting period before breakfast is the longest compared to other meal times in human eating habits. It has been reported that food signal after a long fasting strongly determines the peripheral clock phase [49]. It is also reported that consumption of water-soluble dietary fiber at breakfast which is after a longer fasting period, had greater effects on the microbiota [24]. Soy proteins contain proteins that are resistant to digestion [50], therefore resistant protein becomes the good food for microbiota just like water-soluble dietary fiber. In this study, fasting time was longer before the morning diet than before the evening diet. The intake of soy protein including resistant protein after long fasting may also have a greater impact on the microbiota. Furthermore, it has been reported that microbiota composition has circadian dynamics [21]. Even in mice that were fed the same diet, different reactions may occur if the microbiota is different in the morning or evening. The abdominal temperature, bowel movement, and endocrine system may influence the microbiota diversity [51,52,53,54], and these factors show circadian rhythm [55,56,57]. Thus, the different microbial reactions that were observed based on the soy protein intake in the morning or evening may be explained by the differences in the fasting time before each diet, microbiota circadian oscillations, and gut functional rhythm.

We used PICRUSt analysis to infer the functional capabilities of microbial communities. Soy protein intake upregulated the KEGG pathways associated with amino acid metabolism, especially glycine, serine, and threonine metabolism and lysine biosynthesis (Figure 5a). It has been reported that downregulating the pathway associated with amino acid metabolism has been observed in diarrheic calves and dogs and may be a feature of microbiota-associated diseases [58,59]. It is suggested that soy protein may improve microbiota. The pathways associated with methane and nitrogen metabolisms in the Soy group were significantly upregulated compared to those in the Casein group (Figure 5b). The upregulation of the nitrogen metabolism pathway by soy protein intake suggested that the indigestible component of soy protein might be metabolized by microbiota. The pathways associated with glycine, serine, and threonine metabolism, methane metabolism, and nitrogen metabolism were significantly upregulated in the M-Soy group compared to those in the Casein group (Figure 9). It is suggested that soy protein intake in the morning may have a stronger effect on upregulating these pathways than that in the evening. However, PICRUSt is only a predictor of metagenomic function. Therefore, metabolomic approaches are preferred in identifying factual changes in the metabolic function of microbiota by soy protein intake and its timing and identifying biomarkers for unstable gut microbiota.

## 5. Conclusions

In summary, the present experiments showed that soy protein intake and its timing affected the microbiota. The change in microbiota caused SCFA production and a decrease in cecal pH. In particular, soy protein may be effective in improving lipid metabolism and changing microbiota even with short-term intake. In addition, with respect to the timing of soy protein intake, morning intake may have relatively stronger effects on microbiota than evening intake would. This study provides evidence that soy protein intake and its timing are important factors that affect microbiota composition. To our knowledge, this is the first study to examine the effect of protein intake timing on microbiota and predict the functional profiles of microbial communities affected by soy protein. Therefore, our results are expected to be useful in designing future studies that may focus on the effects of foods or beverages in improving microbiota composition at different mealtimes and in providing important information for chrono-nutrition research.

## Figures and Tables

**Figure 1 nutrients-12-00087-f001:**
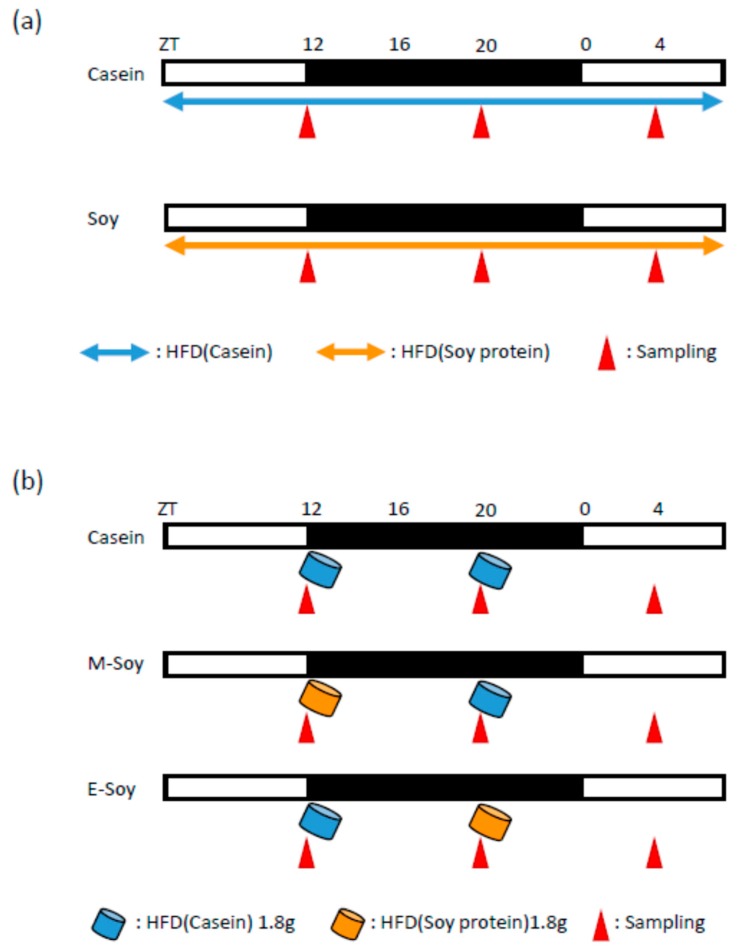
Experimental design. (**a**) Experimental protocol to examine the effect of soy protein intake. (**b**) Experimental protocol to examine the effect of soy protein intake timing. The white and black bars indicate environmental 12 h light and dark conditions, respectively. The horizontal blue arrow indicates free access to a high-fat diet (HFD) with casein. The horizontal orange arrow indicates free access to HFD with soy protein. The blue cylinder indicates the feeding timing of 1.8 g of HFD with casein. The orange cylinder indicates the feeding timing of 1.8 g of HFD with soy protein. The red triangles indicate the sampling time. High-fat diet, casein diet feeding, soy protein diet feeding, morning soy protein diet feeding, evening soy protein diet feeding: HFD, Casein, Soy, M-Soy, E-Soy, respectively.

**Figure 2 nutrients-12-00087-f002:**
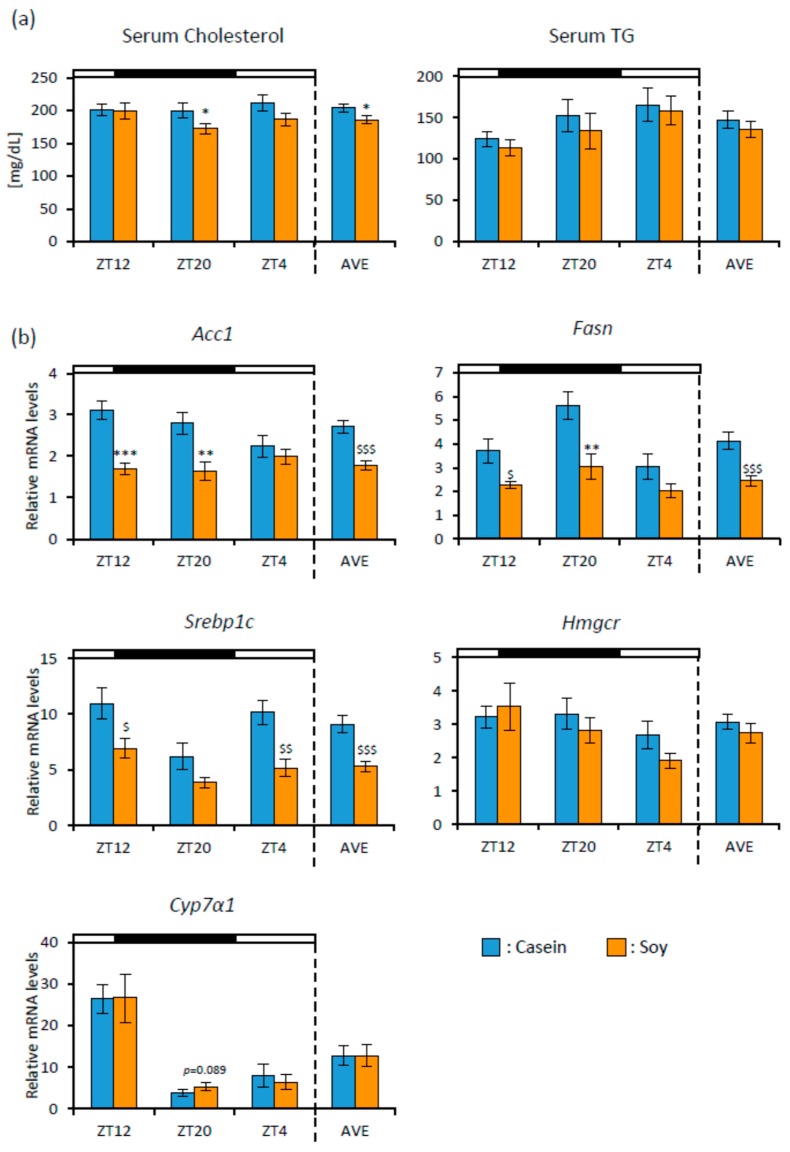
Serum lipid and gene expression levels in the liver. (**a**) Serum cholesterol and triglyceride levels (ZT12, ZT20, ZT4, and an average of three points) of mice that were fed each diet for 10 days. (**b**) Relative RNA expression levels of fatty acid and cholesterol metabolism-related genes in the liver (ZT12, ZT20, ZT4, and an average of three points) of mice that were fed each diet for 10 days. Data are represented as mean ± SEM (*n* = 10). * *p* < 0.05, ** *p* < 0.01, *** *p* < 0.001 versus Casein, evaluated using Student’s *t*-test. $ *p* < 0.05, $$ *p* < 0.01, $$$ *p* < 0.001 versus Casein, evaluated using the Mann-Whitney test. High-fat diet, casein diet feeding, soy protein diet feeding, an average value of three points: HFD, Casein, Soy, AVE, respectively.

**Figure 3 nutrients-12-00087-f003:**
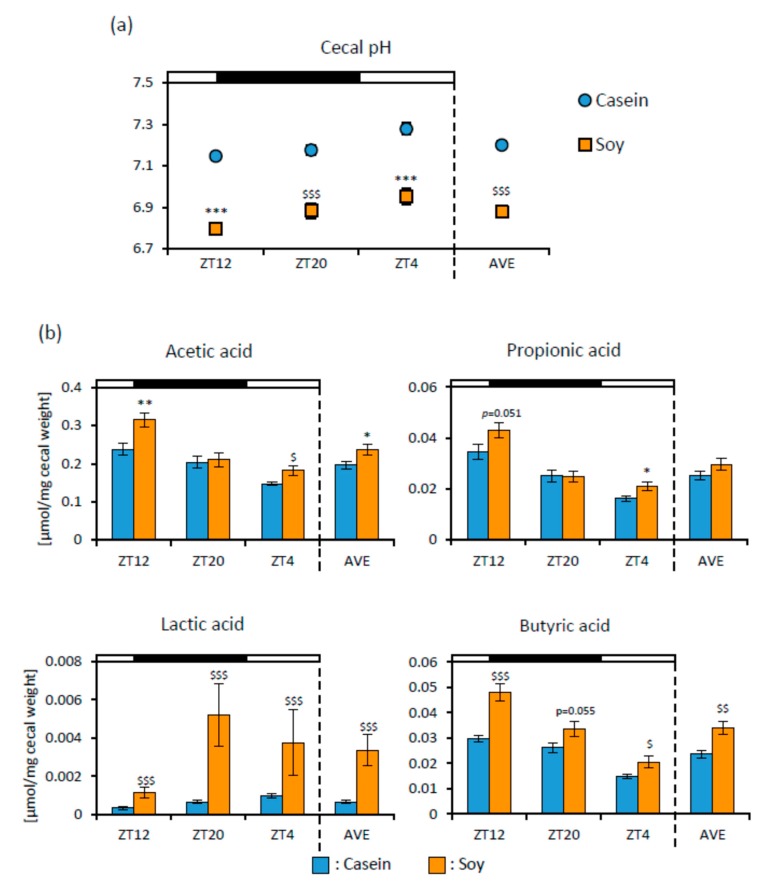
The effect of soy protein intake on cecal pH and SCFA levels. (**a**) Cecal pH levels (ZT12, ZT20, ZT4, and an average of three points) of mice that were fed each diet for 10 days. (**b**) Cecal short-chain fatty acids (SCFA) levels (ZT12, ZT20, ZT4, and an average of three points) of mice that were fed each diet for 10 days. Data are represented as mean ± SEM (*n* = 10). * *p* < 0.05, ** *p* < 0.01, *** *p* < 0.001 versus Casein, evaluated using Student’s *t*-test. $ *p* < 0.05, $$ *p* < 0.01, $$$ *p* < 0.001 versus Casein, evaluated using the Mann-Whitney test.

**Figure 4 nutrients-12-00087-f004:**
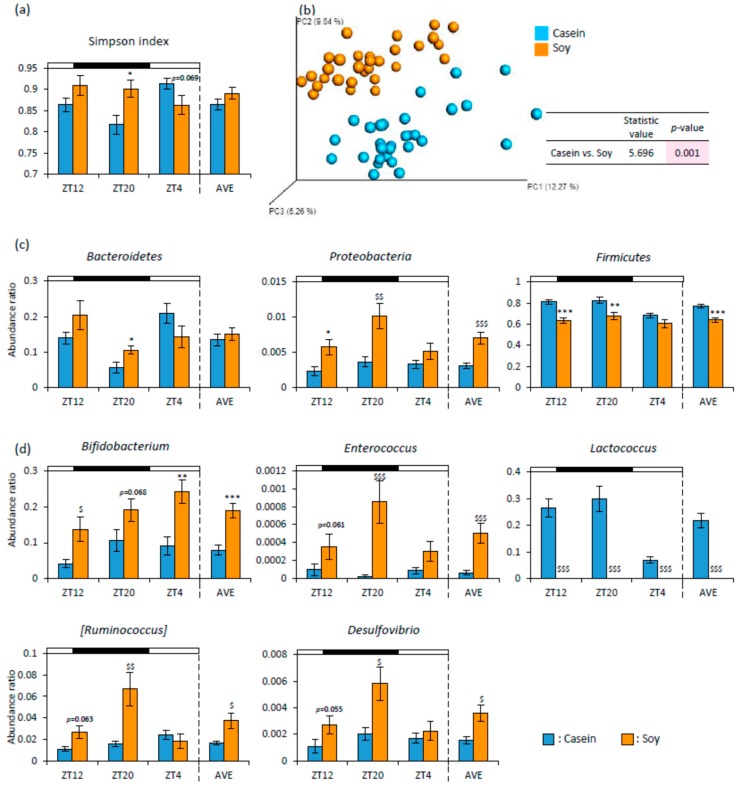
The effect of soy protein intake on microbiota. (**a**) Alpha-diversity about Simpson index (ZT12, ZT20, ZT4, and an average of three points) of mice that were fed each diet for 10 days. (**b**) Beta-diversity in comparison of each diet. The PCoA plots of unweighted UniFrac distance metrics obtained from sequencing the 16S rRNA gene in feces (*n* = 30). (**c**) The relative abundance of microbes at the Phylum level, and (**d**) at the Genus level of mice that were fed each diet for 10 days (ZT12, ZT20, ZT4, and an average of three points). Data (**a**,**c**,**d**) are represented as mean ± SEM (*n* = 10). * *p* < 0.05, ** *p* < 0.01, *** *p* < 0.001 versus Casein, evaluated using Student’s *t*-test. $ *p* < 0.05, $$ *p* < 0.01, $$$ *p* < 0.001 versus Casein, evaluated using the Mann-Whitney test. The table in (**b**) indicates the result using PERMANOVA.

**Figure 5 nutrients-12-00087-f005:**
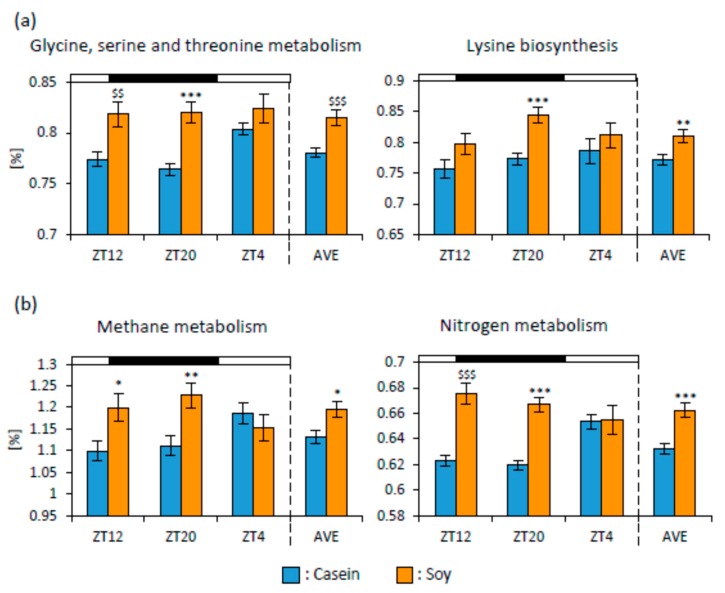
The functional predictions of microbial communities. (**a**) The functional predictions about categories related to “Amino Acid Metabolism” and (**b**) “Energy Metabolism” of microbial communities in mice that were fed each diet for 10 days (ZT12, ZT20, ZT4, and an average of three points). Data are represented as mean ± SEM (*n* = 10). * *p* < 0.05, ** *p* < 0.01, *** *p* < 0.001 versus Casein, evaluated using Student’s *t*-test. $$ *p* < 0.01, $$$ *p* < 0.001 versus Casein, evaluated using the Mann-Whitney test.

**Figure 6 nutrients-12-00087-f006:**
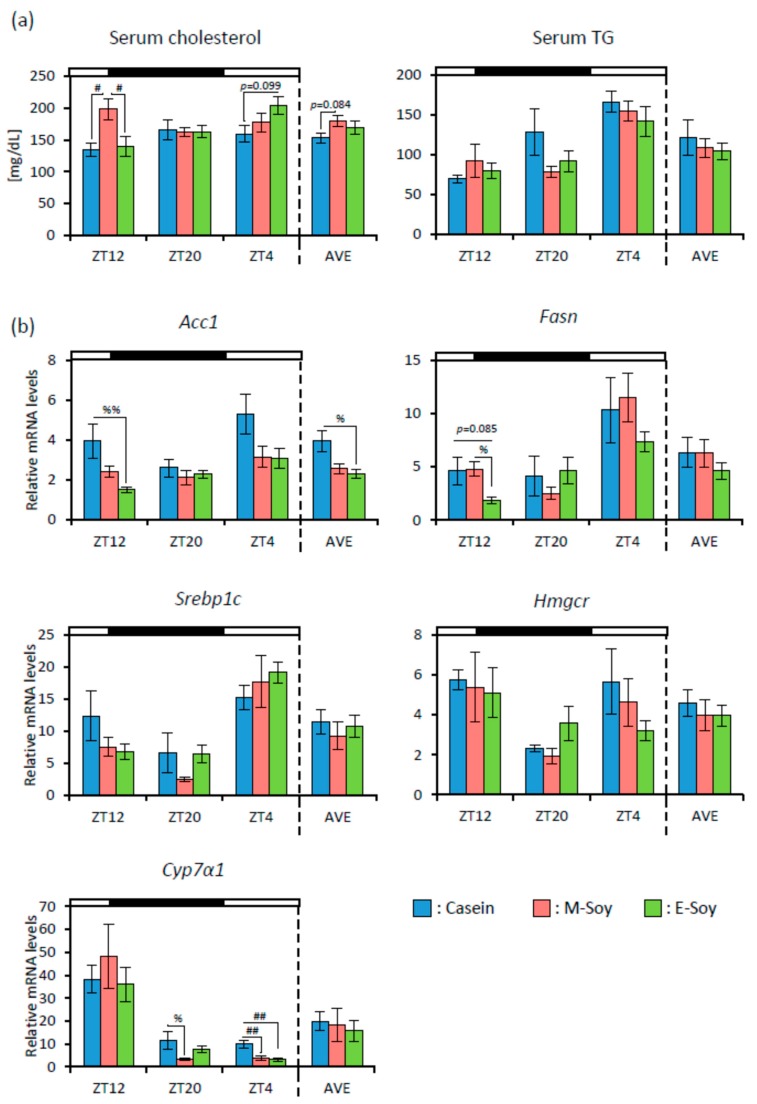
Serum lipid levels and gene expression levels in the liver. (**a**) Serum cholesterol and triglyceride levels (ZT12, ZT20, ZT4, and an average of three points) of mice that were kept in each feeding condition for two weeks. (**b**) Relative RNA expression levels of fatty acid and cholesterol metabolism-related genes in the liver (ZT12, ZT20, ZT4, and an average of three points) of mice that were kept in each feeding condition for two weeks. Data are represented as mean ± SEM (*n* = 5). # *p* < 0.05, ## *p* < 0.01 evaluated using one-way ANOVA with Tukey’s post-hoc test. % *p* < 0.05, %% *p* < 0.01 the Kruskal-Wallis test with Dunn’s post-hoc test. High-fat diet, casein diet feeding, morning soy protein diet feeding, evening soy protein diet feeding, the average value of three points: HFD, Casein, M-Soy, E-Soy, AVE, respectively.

**Figure 7 nutrients-12-00087-f007:**
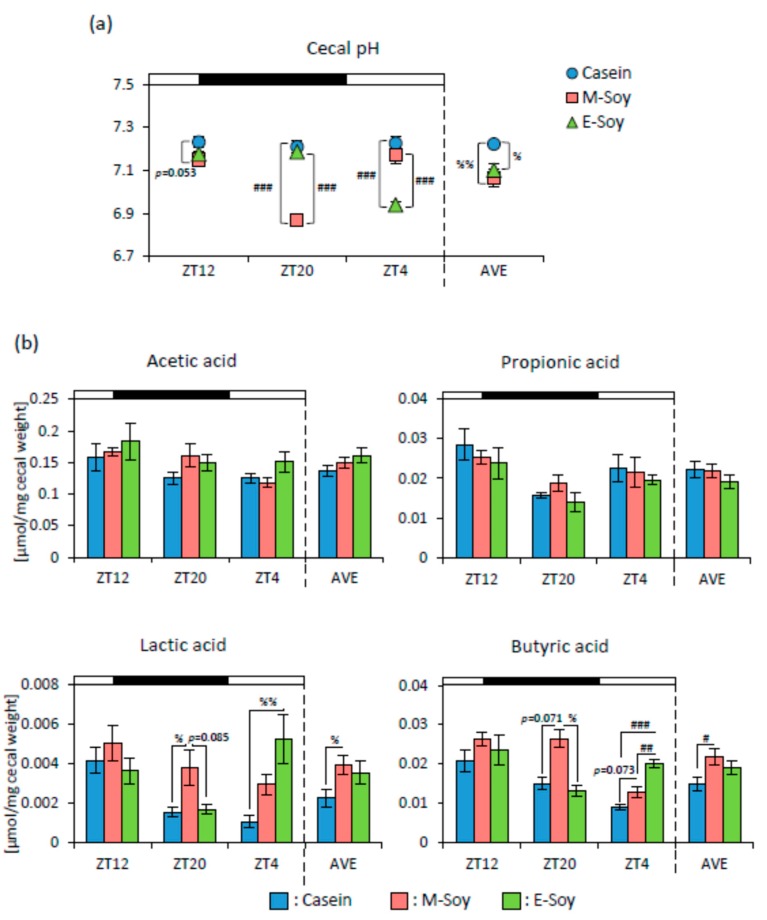
The effect of soy protein intake timing on cecal pH and SCFA levels. (**a**) Cecal pH levels (ZT12, ZT20, ZT4, and an average of three points) of mice that were kept in each feeding condition for two weeks. (**b**) Cecal SCFA levels (ZT12, ZT20, ZT4 and average of three points) of mice that were kept in each feeding condition for two weeks. Data are represented as mean ± SEM (*n* = 5). # *p* < 0.05, ## *p* < 0.01, ### *p* < 0.001 evaluated using one-way ANOVA with Tukey’s post-hoc test. % *p* < 0.05, %% *p* < 0.01 the Kruskal-Wallis test with Dunn’s post-hoc test.

**Figure 8 nutrients-12-00087-f008:**
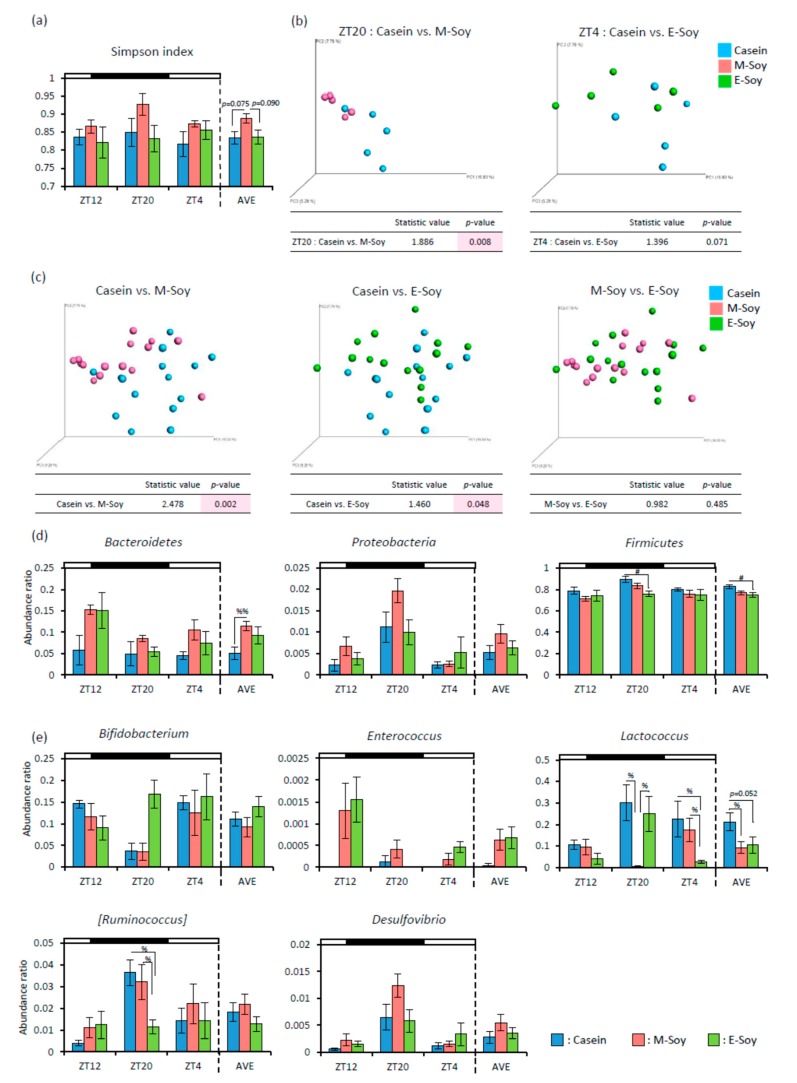
The effect of soy protein intake timing on microbiota. (**a**) Alpha-diversity about Simpson index (ZT12, ZT20, ZT4, and an average of three points) of mice that were kept in each feeding condition for two weeks. (**b**) Beta-diversity in comparison of Casein and M-Soy at ZT20, or Casein and E-Soy at ZT4, shortly after soy protein intake (*n* = 5). (**c**) Beta-diversity in comparison of Casein and M-Soy, Casein and E-Soy, or M-Soy and E-Soy. The PCoA plots of unweighted UniFrac distance metrics obtained from sequencing the 16S rRNA gene in feces (*n* = 15). (**d**) The relative abundance of microbes at the Phylum level, and (**e**) at the Genus level of mice that were kept in each feeding condition for two weeks. Data (**a**,**d**,**e**) are represented as mean ± SEM (*n* = 5). # *p* < 0.05 evaluated using one-way ANOVA with Tukey’s post-hoc test. % *p* < 0.05, %% *p* < 0.01 the Kruskal-Wallis test with Dunn’s post-hoc test. The tables in (**b**,**c**) indicate the result using PERMANOVA.

**Figure 9 nutrients-12-00087-f009:**
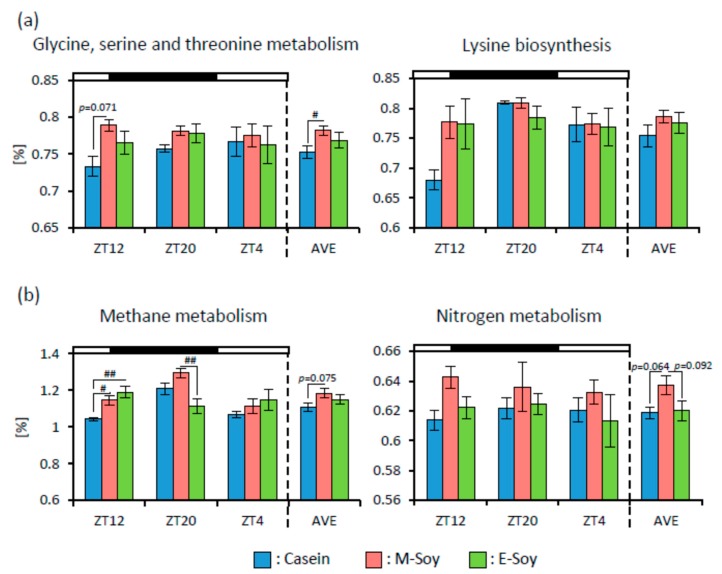
The functional predictions of microbial communities. (**a**) The functional predictions about categories related to “Amino Acid Metabolism” and (**b**) “Energy Metabolism” of microbial communities in mice that were kept under each feeding condition for two weeks (ZT12, ZT20, ZT4, and an average of three points). Data are represented as mean ± SEM (*n* = 5). # *p* < 0.05, ## *p* < 0.01 evaluated using one-way ANOVA with Tukey’s post-hoc test.

**Table 1 nutrients-12-00087-t001:** Nutrition components (g) in each diet (100 g).

	Casein Diet	Soy Diet
Casein	22.86	-
Soy protein	-	23.78
L-cysteine	0.18	0.18
βcorn starch	13.71	12.79
αcorn starch	15.5	15.5
Sucrose	10	10
Soybean oil	4	4
Lard	24	24
Cellulose	5	5
Mineral mixture	3.5	3.5
Vitamin mixture	1	1
Choline bitartrate	0.25	0.25
Total	100	100

**Table 2 nutrients-12-00087-t002:** Sequences of Primers for Real-time RT-PCR.

Gene	Forward	Reverse
*Acc1*	GCACTCCCGATTCATAATG	CCCAAATCAGAAAGTGTATC
*Cyp7* *α1*	AGACCGCACATAAAGCCCGG	CTTTCATTGCTTCAGGGCTC
*Fas*	TGGGTTCTAGCCAGCAGAGT	ACCACCAGAGACCGTTATGC
*Gapdh*	TGGTGAAGGTCGGTGTGAAC	AATGAAGGGGTCGTTGATGG
*Hmgcr*	GATCATCCAGTTGGTCAATGC	GCAAGCTTTGTGGAGAGGAG
*Srebp1c*	CGCTACCGGTCTTCTATCAATG	CAAGAAGCGGATGTAG

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
