# Peer review of "The Timing Effects of Soy Protein Intake on Mice Gut Microbiota"

_nutrients, 2019, doi:10.3390/nu12010087_

Round 1
Reviewer 1 Report
The manuscript entitled "The timing effects of soy protein intake on mice gut microbiota" by Konomi Tamura, et al. In this study, the authors found that soy protein intake had an impact on the lipid metabolism and microbiota changes in mice guts The subject and aim of this study would be of interest for readers of Nutrients, however, the current form should be revised in some points.
In Materials and Methods section, a detailed description of the work-flow of the two experiments is missing. From the second section I could not find out that two types of experiments have been carried out. There no explanation about scheme of mice feeding in two kind experiments. This information is mentioned only in "Results" but it should be described in the previous section. Material and method section lacks:
- Amounts of mice used in both experiments, how many mice were in one cage?
- A detailed description of two experiments
- Time of mice sacrifice and gene expression measurements. In section 2.3 there no information that mice were sacrificed in this time points (ZT12, ZT10, ZT4). It seems that all were anesthetized in a one-time point for a liver collection. Please describe in detail the process of anesthetizing, time od sampling for all measurements in all subsections.
- -Description of the cecal collection (is dependent on if mice were in a group in one cage or were housed individually one mice per cage.
Did the authors evaluate the weight of obese mice before and after the experiment. How administration of soy protein affected the weight of mice. Is the supply of soy protein could help to combat obesity in humans?
Discussion section should be improved. The authors repeated results described in previous section. Obtained results are poorly discussed and cited references are described very general (e.g. lines 374, 383) Some of the authors' suppositions are not supported by references.
The manuscript needs major revision.
Reviewer 2 Report
This study investigated the effects on gut microbiota of a soy protein diet administered to ICR 8 week-old male mice under a previous high fat diet (HFD) condition. Additionally, the morning vs evening effects of soy protein diet were evaluated in relation to a casein protein diet that was used as control treatment. The authors conclude that as soy protein intake results in higher short-chain fatty acid (SCFAs) production, which is associated to decreased cecal pH and increased microbiota diversity. This effect was greater when soy protein feeding was allowed in the morning than in the evening, suggesting that morning intake may have stronger effects on microbiota than that in the evening. The study design and methodology seems to be solid although I have some concerns. Additionally, it is not clear to what extent the differences between morning and afternoon intake were due to different fasting times in both groups of mice, since this could lead to a different metabolic status, then arising different responses at the microbiota level. It is also suggested that the daily oscillations in the microbiota could play some role in the morning-evening differences, although the study does not provide evidence about it. However, the results of the study are in general relevant and the conclusions help to understand the potential benefit of the soy protein intake.
I have some doubts and comments that the authors should clarify.
Introduction. The section is clear and well organized. However, the penultimate paragraph is quite confusing. On the one hand, a relationship of the intestinal microbiota with the circadian system is established and on the other hand, the existence of diurnal changes in the composition and function of the microbiota and its function is appointed. However, it is not apparent how such changes can be associated with the intake and influenced by diet composition (soy protein). The text perhaps would be improved if the underlying role that intake timing and food composition may have in the microbiota is better focused.
M&M. Information concerning the experimental design and sampling that appears in this section is incomplete. An important amount of experimental data appears in the results section, which is confusing for a reader. I suggest the authors to include at least the more relevant details in the material and methods section.
Apparently, the amount of food fed by mice in experiment 1 (free access to food) was higher than that given to mice in experiment 2 following a manual feeding (twice a day, 1.8 g HSD with soy protein given in the morning or in the evening). Methodologically this may be a disappointment. I have several questions regarding the feeding strategy. Were the different diets –different protein composition- similarly accepted by mice? In experiment 1, was the total food intake quantified? It would be important since a similar food amount could have being given to mice in experiment 2. Why was the manual feeding chosen for experiment 2 instead of providing a short-time period (i.e., two hours) of access to food in the morning or evening?
It is not evident the reason for choosing a different duration of the soy protein diet treatments in both experiments, that is, 10 days to see the effects of the diet on the lipid metabolism and the microbiota and two weeks to see the timing effects (morning-evening). Additionally, it is unclear why a short time feeding strategy was chosen instead of a longer-term application for which there are previous studies that could serve as a guide for the current study.
Although the changes induced by the administration of the soy diet in lipid metabolism are quite clear at the level of the expression of fatty acid synthesis related genes, it would be interesting to include data of the activity of some enzymes (ACL, ACO-AC, FAS,..).
Timing effects. Apparently, there could be a relationship between the effects observed in the microbiota at each time of the day (morning and evening) with the elapsed time from the feeding to sampling. For example, as shown in Figure 6, the decrease in cecal pH and the increase in the cecal SCFA levels observed with the morning soy intake occurred only in the sampling time closest to the feeding period, that is, ZT20. Similarly, with the evening soy diet, the effect of lowering cecal pH and increasing lactic acid and butyric acid levels occurred only at ZT4, being that the sampling point closest to the time at which the animal had access to food. To me this data suggests that there could be an important influence of the metabolic status related to previous feeding, or to hormonal regulation associated with intake. Perhaps the authors could add some discussion about it.
The authors conclude that two weeks of soy protein feeding in the morning or evening resulted in a stronger decrease in cecal pH and increased SCFA levels in mice fed in the morning compared to those fed in the evening. However, when looking to Figure 6 this is unclear. Apparently, the soy diet given during the evening produced similar effects on the decrease in cecal pH and the increase in SCFA levels than that given during the morning, which would indicate similar general effects on the microbiota. My question is about the possibility to draw the same conclusion from diets given in the evening than in the morning. On the other hand, subsequent analysis show that the morning diet has a greater impact on the abundance and diversity of the microbiota. However, regarding the latter the greater effect of morning diet in the presence of positive bacteria for intestinal function is unclear. At which level are the obtained data conclusive in terms of benefits of soy intake at morning vs evening for the gut function?
Ln- 387-389. Are the authors suggesting an implication of GLP-1 on the effects observed in this study? If so, the text should be clarified and justified.
Ln. 417-418. The authors should be cautios when doing comparisons between the effects of restricted feeding and the feeding strategy followed in the current study.
Ln. 446. ..fating. It should be fasting.
Round 2
Reviewer 1 Report
Thank you for the improved version of your manuscript. I accept it in current version.